Herpetogaster collinsi from the Cambrian of China elucidates the dispersal and palaeogeographic distribution of early deuterostomes and the origin of the ambulacrarian larva

Yang Xianfeng 1 2 3 yangxf@ynu.edu.cn
http://orcid.org/0000-0001-8032-4272 Kimmig Julien 4 5 jkimmig@psu.edu
http://orcid.org/0000-0003-4726-0355 Schiffbauer James D. 6 7
Peng Shanchi 3
1 Yunnan Key Laboratory for Palaeobiology, Yunnan University , Kunming , China
2 MEC International Joint Laboratory for Palaeobiology and Palaeoenvironment, Yunnan University , Kunming , China
3 State Key Laboratory of Palaeobiology and Stratigraphy, Chinese Academy of Sciences , Nanjing , China
4 Paläontologie und Evolutionsforschung, Abteilung Geowissenschaften, Staatliches Museum für Naturkunde Karlsruhe , Karlsruhe , Germany
5 The Harold Hamm School of Geology & Geological Engineering, University of North Dakota , Grand Forks, North Dakota , United States
6 Department of Geological Sciences, University of Missouri-Columbia , Columbia, Missouri , United States
7 X-ray Microanalysis Laboratory, University of Missouri-Columbia , Columbia, Missouri , United States
Bona Paula
Electronic publication date: 2023 Nov 7
Publication date: 2023
Volume: 11
Electronic Location ID: e16385
Received 2023 Jul 6; Accepted 2023 Oct 10
Copyright: © 2023 Yang et al.
Copyright year: 2023
Copyright holder: Yang et al.
License: This is an open access article distributed under the terms of the Creative Commons Attribution License, which permits unrestricted use, distribution, reproduction and adaptation in any medium and for any purpose provided that it is properly attributed. For attribution, the original author(s), title, publication source (PeerJ) and either DOI or URL of the article must be cited.
License URL: https://creativecommons.org/licenses/by/4.0/

Keywords: Deuterostomia, Ambulacraria, Herpetogaster, Lifestyle, Dispersal, Cambrian Stage 4, Exceptional preservation, larvae, Palaeoecology, Palaeogeography

Funding: National Natural Science Foundation of China 42162001, 41062001 and 41562001 Nanjing Institute of Geology and Palaeontology, CAS 103113 NSF CAREER 1652351 Financial support was provided by the National Natural Science Foundation of China (Grant Nos. 42162001, 41062001, 41562001) and the State Key Laboratory of Palaeobiology and Stratigraphy (Nanjing Institute of Geology and Palaeontology, CAS) (Grant No. 103113) to Xianfeng Yang. James D Schiffbauer is supported by NSF CAREER 1652351. The funders had no role in study design, data collection and analysis, decision to publish, or preparation of the manuscript.

==============================
The Cambrian Radiation represents one of the largest diversification events in Earth history. While the resulting taxonomic diversity is exceptional, relatively few of these novel species can be traced outside the boundaries of a single palaeocontinent. Many of those species with cosmopolitan distributions were likely active swimmers, presenting opportunity and means to conquer new areas, but this would not have been the case for sessile organisms. Herpetogaster is a lower to middle Cambrian (Series 2–Miaolingian, Stage 3–Wuliuan) genus of sessile, stalked, filter-feeding deuterostomes with two species, H. collinsi and H. haiyanensis, known respectively from Laurentia and Gondwana. Here, we expand the distribution of H. collinsi to Gondwana with newly discovered specimens from the Balang Formation of Hunan, China. This discovery raises questions on the origin of the genus and how sessile organisms were able to disperse over such a broad distance in the lower Cambrian. As Herpetogaster has been recovered at the base of the Ambulacrarian tree in recent phylogenies, a planktonic larval stage is suggested, which implies, that the last common ancestor of the Ambulacraria might have already had planktonic larvae or that such larvae developed multiple times within the Ambulacraria.

Introduction

During the Cambrian Radiation, hundreds of metazoan species, representing nearly all known modern phyla, appear in the fossil record and can be found throughout rich fossil deposits around the world (Paterson et al., 2016; Fu et al., 2019; Harper et al., 2019; Kimmig et al., 2019; Nanglu, Caron & Gaines, 2020; Yang et al., 2021). While there is an excellent record of soft-bodied fossils from globally distributed Burgess Shale-type Lagerstätten as early as Cambrian Stage 3 (~521–514 Ma, not formally ratified), few species can be found crossing palaeocontinental boundaries (Paterson et al., 2016; Fu et al., 2019; Harper et al., 2019; Kimmig et al., 2019; Nanglu, Caron & Gaines, 2020; Saleh et al., 2020; Yang et al., 2021). This pattern of species distribution not only begs important questions about modes of reproduction and broader-scale mobility or migration of species in the lower Cambrian, but also how the few species with cosmopolitan distributions were able to accomplish this feat. Based on newly discovered fossils presented herein, one species that had successfully trekked and established a foothold on multiple palaeocontinents is Herpetogaster collinsi Caron, Conway Morris & Shu, 2010.

Herpetogaster, one of the earliest-known representatives of the invertebrate deuterostomes, is a stalked filter feeder grouped with the informal cambroernid clade. The cambroernids are a group of fossils, comprising two distinct body plans (Caron, Conway Morris & Shu, 2010; Nanglu et al., 2023). The first is represented by a group of discoidal fossils with coiled coelomic sac and dendritic feeding tentacles, the eldonioids. The other morphology is that of Herpetogaster. The genus is represented by two species found in the lower to middle Cambrian (Series 2–Miaolingian, Stage 3–Wuliuan) of Gondwana and Laurentia (Caron, Conway Morris & Shu, 2010; Kimmig, Meyer & Lieberman, 2019; Yang et al., 2020; Nanglu et al., 2023; Pari, Briggs & Gaines, 2022). To date, its two species have been observed to be restricted to their respective palaeocontinents: H. haiyanensis to Series 2 deposits in Gondwana and H. collinsi to generally younger Miaolingian deposits in Laurentia (with the exception of the Series 2 Pioche Formation). From all known fossils of the genus, Herpetogaster appears to have been a sessile organism, with a stolon and a basal discoidal holdfast. While the stolon may have been contractible, its likely function was to anchor the organism in place within the sediment. Neither of the two species preserve swimming appendages, and thus the presumed mechanism of expanding its distribution would have been through passive transport by ocean currents. Recent flume experiments have shown that soft-bodied animals can hypothetically be transported over tens of kilometers by current flow (Bath Enright et al., 2021), but this is an improbable explanation for the occurrence of the same species separated by thousands of kilometers and within deposits of different ages. The Laurentian and South China paleocontinents were not only separated by a vast distance but also by other continents, e.g., Siberia and Australia, on each side. These continental obstacles would have needed to be circumvented or Herpetogaster larvae could have settled in these locations as intermediate steps. With specimens of H. collinsi reported here from the Balang Formation, Hunan, China, along with possible larval fossils, we propose that these organisms possessed a motile larval stage that provided the means for broad dispersal and migration.

Thus far, we have collected more than 60 new specimens of H. collinsi from the lower Cambrian (Series 2, Stage 4) Balang Formation. Here, we provide detail of 19 of the best-preserved new specimens, explore the distribution of the genus in time and space, and postulate mechanisms that might have led to its success.

Materials and Methods

The 19 new specimens of Herpetogaster collinsi described here are reposited in the collections of the Yunnan Key Laboratory for Palaeobiology, Yunnan University, Kunming, China (YKLP), with specimen numbers YKLP 14,570–14,588. Specimens with the prefix ROM are housed in the Royal Ontario Museum, Ontario, Canada, with the prefix KUMIP at the Division of Invertebrate Paleontology, Biodiversity Institute, University of Kansas, and with the prefix YPM at the Yale Peabody Museum of Natural History.

Imaging

Pictures of the specimens were collected as previously described in Yang et al. (2020, 2021, 2022). Specifically the specimens were photographed using a Canon EOS 5D digital SLR camera with a Canon 50 mm macro lens and cross-polarized lighting. Close-ups were captured using a Leica DFC 500 digital camera mounted on a Leica M205-C stereoscope. All specimens were submerged in alcohol to increase contrast. The contrast, colour space, and brightness were adjusted using Adobe Photoshop CC.

Scanning electron microscopy and energy-dispersive X-ray spectroscopy analyses

The backscattered electron (BSE) imaging and energy-dispersive X-ray spectroscopy (EDS) of uncoated specimens was conducted with a FEI Quanta 650 FEG field emission scanning electron microscope (SEM) at the Yunnan Key Laboratory for Palaeobiology, Institute of Palaeontology, Yunnan University, Kunming, China (YKLP). Imaging of the specimens was done as previously described in Yang et al. (2020, 2021, 2022). Specifically, all imaging analyses were conducted with the following operating conditions: 14 ± 1 mm working distance (minor differences to allow for variation in sample thickness or topography) for basic imaging and EDS, 20 keV beam accelerating voltage, 10 nA beam current, 20 Pa chamber pressure (low vacuum), 50 µm aperture for imaging, and 40 µm aperture for EDS analysis. The EDS analyses were repeated with a 10 keV beam accelerating voltage to lessen beam penetration depth; the results are available in the Supplemental Material.

Geological setting

The specimens described herein were collected from the lower part of the Balang Formation in northwestern Hunan, China (Figs. 1A–1C). The Balang Formation is part of a conformable lower Cambrian sequence, positioned between the Niutitang Formation below and the Chinghsutung Formation above (Fig. 1B). The presence of the trilobite Oryctocarella duyunensis confirms that the collection interval is positioned within Series 2, Stage 4 of the Cambrian System (Peng et al., 2017; National Commission on Stratigraphy of China, 2018; Zhao et al., 2019; Dai et al., 2021). In the context of other well-known Chinese Cambrian Lagerstätten, the Balang biota is situated in age between the slightly older Chengjiang biota (Series 2, Stage 3) and younger Kaili biota (Miaolingian, Wuliuan). The sampled outcrop, located approximately 32 km south-west of Huayuan town, is composed of finely laminated dark grey calcareous mudstone intervals, interbedded with silty mudstone and silty shale (Fig. 1B).

Figure 1 Distribution of Herpetogaster.

(A) Location of the studied section of the Balang Formation located approximately 32 km south-west of Huayuan town, Hunan Province, South China. (B) Generalized stratigraphy of the Balang Formation at this location. (C) Palaeogeographical distribution of Herpetogaster specimens during Cambrian Stages 3–4 and during the Wuliuan (map modified from Streng & Geyer (2019)). 1. H. hanyanensis (Chengjiang biota, Yunnan, China, Stage 3), 2. H. collinsi (Balang biota, Hunan, China, Stage 4), 3. H. collinsi (Burgess Shale, British Columbia, Canada, Wuliuan), 4. H. collinsi (Pioche Formation, Nevada, USA, Stage 4), 5. H. sp. (Parker Formation, Vermont, USA, Stage 4). Abbreviations: Ni., Niutitang Formation; Chi, Chinghsutung Formation.

Results

Systematic Paleontology

Superphylum: Deuterostomia Grobben, 1908

Clade: Ambulacraria Metschnikoff, 1881

Unranked stem-group: Cambroernida Caron, Conway Morris & Shu, 2010

Genus Herpetogaster Caron, Conway Morris & Shu, 2010

Herpetogaster collinsi Caron, Conway Morris & Shu, 2010

Holotype. ROM 58051

New Material. YKLP 14570–14588

Provenance. Balang Formation, lower Cambrian (Series 2, Stage 4), Oryctocarella duyunensis biozone, Mozi village, Paiwu township, approximately 32 km south-west of Huayuan town, Hunan Province, China. Pioche Formation, Comet Shale Member; lower Cambrian (Series 2, Stage 4), Nephrolenellus multinodus biozone; Ruin Wash, NW ¼ SW ¼ sec. 15, R65E T2S, 17 km west of Panaca, Lincoln County, Nevada (see Palmer, 1998, and Lieberman, 2003 for greater discussion of the locality). Burgess Shale and Stephen Shale Formations; middle Cambrian (Miaolingian, Wuliuan); Yoho and Kootenay National Parks, British Columbia, Canada.

Diagnosis. Segmented body, coiled dextrally. Short head bearing prominent bilateral anterior dendritic tentacles of sub-equal length and in two-by-two arrangement with pharyngeal structures, possibly lateral pores. Trunk subcylindrical, divided into two subsections, narrowing posteriorly. Ventral and contractile adhesive stolon, sometimes with terminal disc. Digestive tract with anterior mouth, pharynx, voluminous stomach, and narrow intestine with terminal anus. Stomach and intestine of sub-equal lengths, un-looped, with triangular mesenterial insertions (from Caron, Conway Morris & Shu, 2010).

Description. Most of the specimens are complete and range in length from 8.4 to 53.6 mm and from 2.3 to 7.9 mm in width. The body is curved, some specimens show fine segmentation along the trunk (Figs. 2C, 2E, 2G, 2J, 3C and 3F), and ends in an anus (Figs. 2D–3H, 3J, 3C and 3E–3G).

Figure 2 Complete specimens of Herpetogaster collinsi Caron, Conway Morris & Shu, 2010 from the Balang Formation of China.

(A) YKLP 14570 a small, likely juvenile specimen, preserving slim paired tentacles and the extended stolon. (B and C) YKLP 14571 and YKLP 14572 part and counterpart of a likely sub-adult specimen, the trunk is almost completely preserved as a black carbonaceous film, with the detail of dendritic symmetrical tentacle and a stolon. (D) YKLP 14573, a large adult specimen, with the showing the branching tentacles. Co-occurring with a relative smaller, maybe sub-adult individual. (E) YKLP 14574, adult specimen with preserved digestive tract, prominent segments and segment boundaries (indicated by arrows). (F) YKLP 14575, adult specimen coiled with tentacles, part of the digestive tract, and stolon. (G) YKLP 14576, adult specimen with tentacles, well-preserved anus and terminal disc. (H) YKLP 14577, adult specimen with both tentacles and well-preserved anus. (I) YKLP 14578, sub-adult specimen with well-preserved branching tentacle, stolon and terminal disc. (J) YKLP 14579, adult specimen with showing tentacles, the digestive tract, anus, segments and segment boundaries. Scale bars: (A) 2 mm; (B–J) 5 mm. Abbreviations: an, anus; in, intestine; p?, putative pharyngeal pores; ph, pharynx; seg, segment boundary?; st, stolon; stom, stomach; td, terminal disc; te, tentacle.

Figure 3 Gregarious specimens of Herpetogaster collinsi from the Balang Formation of China.

(A) YKLP 14580, large adult specimen is found co-occurring with associated plankton, possibly larvae. (B) YKLP 14581, close-up of the associated tornaria-like structure, which shows similarities to extant early stage ambrulacarian larvae. (C) YKLP 14582, at least six specimens of H. collinsi preserved on a single slab. (D) YKLP 14584, two adult specimens with almost complete paired tentacles, one specimen preserved the stolon. (E) YKLP 14585, two adult specimens preserving the tentacles, digestive tract, and anus. (F) YKLP 14583, counterpart of (C). (G) YKLP 14586, two complete adult specimens preserving tentacles, anus and the stolon. (H) YKLP 14587, three specimens with almost complete paired tentacles, one preserving the stolon and terminal disc. (I) YKLP 14588, at least two adult specimens preserving the stolon and terminal disc. Scale bars: (B) 1 mm; (A, C–I) 5 mm. Abbreviations: an, anusph, pharynx; seg, segment boundary?; st, stolon; td, terminal disc; te, tentacle.

The head has a maximum width of 7.4 mm and a maximum length of 3.9 mm. The pharynx is visible in nine specimens. Two tentacles emerge from the corners of the head, they reach up to 14.3 mm in length, bifurcate at 0.6 to 5 mm and preserve between 10 and 11 branches.

The trunk ranges from 4.8 to 36.1 mm in length, which equates to about three-quarters of the total body length of the animal. The other quarter represents the pharynx, 0.7 to 3.6 mm in length. Some specimens (n = 9; Figs. 2E, 2G, 2I, 2J, 3A–3C, 3E, 3F and 3I) preserve a darker internal structure, which is interpreted as the stomach and digestive tract as it reaches the anus in YXLP 14574 and YKLP14579 (Figs. 2E and 2J). The soft-tissue surrounding the stomach and intestine preserves fine segmentation (Figs. 2E, 2G, 2J, 3C and 3F), separating the trunk into 13 segments.

The stolon extends from the final third of the trunk, around the ninth or tenth segment (Figs. 2E, 2J, 3C and 3F), and varies in length (1.5 to 14.3 mm), width (0.4 to 1.8 mm), and width:length ratio (0.06 to 0.50). This suggests that the stolon may have been contractible. Some specimens preserve a terminal disk at the end of the stolon, interpreted as the holdfast (Figs. 2G, 2I, 3C, 3H and 3I).

Preservation. Balang Formation specimens are usually regarded as carbonaceous compressions preserved within carbonaceous mudstones, similar to those of other Cambrian Burgess Shale-type deposits (e.g., Wen et al., 2019). Two specimens of Herpetogaster collinsi were analysed using SEM-EDS (Fig. 4). The first specimen (YKLP 14583) has a distinctive, if not continuous carbon signature (Fig. 4L), whereas the other specimen (YKLP 14573) did not provide a strong carbon signature, but a more continuous film (Fig. 4E). The analyses also show enrichment of iron and phosphate in the host rock, though neither elemental signature appears to be associated with the fossils themselves. This suggests that that diagenesis played a vital role in the preservation of carbon in the Balang Formation, but also suggests that all the Herpetogaster specimens underwent the same taphonomic process. As observed by other authors (Broce & Schiffbauer, 2017; Leibach et al., 2021) in specimens from the Marjum Formation of Utah, the carbon signature in YKLP 14573 correlates spatially with calcium (Fig. 4C). This suggests that some of the signal is likely derived from the host rock. This correlation is visible with 10 keV and 20 keV beam acceleration voltage. However, these elemental signatures do not correlate in YKLP 14583. The previous studies inferred that the higher carbon concentration observed locally stemmed from accumulation of carbonaceous material within interstitial spaces between grains of the host rock (Broce & Schiffbauer, 2017; Leibach et al., 2021). The absence of detectable carbonaceous films or local enrichments of carbon in YKLP 14573 might indicate a few possible taphonomic scenarios involving the complete removal of organic carbon. For example, given their infaunal nature, the organisms likely died within the uppermost oxic- to sub-oxic layers of sediment, allowing for efficient decay by oxygenic microbes. Burial occurred rapidly after death, as indicated by their mostly complete preservation and the presence of preserved labile structures (e.g., tentacles, stolon, gut). Organic carbon removal/dispersion could have continued through diagenesis (Leibach et al., 2021) as well or modern episodes of weathering, such as precipitation, could have leached organic material (Saleh et al., 2021; Whitaker et al., 2022).

Figure 4 SEM micrograph and SEM-EDS elemental maps of Herpetogaster collinsi from the Balang Formation of China.

(A–G) YKLP 14573. (A) Picture of the specimen indicating the analyzed area (dashed rectangle). (B) Detailed view of the analyzed area. (C–G) SEM-EDS elemental maps of Ca, Fe, C, P, S, respectively. (H–N) YKLP 14583. (H) Picture of the specimen indicating the analyzed area (dashed rectangle). (I) Detailed view of the analyzed area. (J–N) SEM-EDS elemental maps of Ca, Fe, C, P, S, respectively. Scale bars: (A, H) 5 mm; (C–G, J–N) 1 mm.

An interesting feature of the Balang specimens is that several have a relatively uniform preservational coloration (Figs. 2A, 2C, 2D, 3G and 3H) as compared to representatives of the genus in other deposits. This may be a result of additional decomposition of the specimens after burial, leading to decay of the internal organs, and a generally more homogeneous, black-film appearance. Additionally, rather than all dendritic in form, the tentacles are preserved in different morphologies, as opposed to the Burgess Shale and Pioche Formation specimens (Fig. 5). Enhanced decomposition may again be the culprit for this lack of dendritic detail; overlapping tentacles may also reduce the appearance of finer details, though this is not expected to be the case in all the examined Balang specimens.

Figure 5 All known Herpetogaster species from Gondwana and Laurentia.

(A) YKLP 14404, holotype of Herpetogaster haiyanensis from the Chengjiang biota (Cambrian; Series 2; Stage 3) of Yunnan, China; (B) YPM IP 239054, ?Herpetogaster sp. from the Parker Formation (Cambrian; Series 2; Stage 4) of Vermont, USA; (C) YKLP 14576, Herpetogaster collinsi from the Balang Formation (Cambrian; Series 2; Stage 4) of China; (D) KUMIP 482878, Herpetogaster collinsi from the Pioche Formation (Cambrian; Series 2; Stage 4) of Nevada, USA; (E) ROM 58051, holotype of Herpetogaster collinsi from the Burgess Shale (Cambrian; Miaolingian; Wuliuan) of British Columbia, Canada. Scale bars: 5 mm.

Remarks. The Balang Formation specimens are assigned to Herpetogaster collinsi. Though there are taphonomic differences as described above, these specimens appear to be nearly morphologically identical to those described from the Burgess Shale (Caron, Conway Morris & Shu, 2010) and the Pioche Formation (Kimmig, Meyer & Lieberman, 2019). They preserve the dextrally coiled body, occupying about three-quarters of the animals’ total body length; anterior dendritic tentacles that emerge from the corner of the head and bifurcate; pharyngeal structures; a digestive tract; and a stolon but with no apparent division into inner and outer layers or terminal disc. Segmentation is also prominent in some of the Balang specimens (Figs. 2D, 2E, 2G, 2H, 2J, 3A, 3C, 3E and 3F), with segmentation lines visible in 9 of 19 closely examined specimens.

These Balang specimens visibly differ from H. haiyanensis, which has over one hundred branches per tentacle, a single layered stolon, and preserved inner and outer layers (Yang et al., 2020)—none of which can be explained by taphonomic differentiation though were plausibly beneficial adaptations to the deltaic environment which they occupied (Peng, 2009; Saleh et al., 2022).

The Balang specimens appear to have been gregarious or living in close proximity; up to six specimens have been found on a single slab (Figs. 3C and 3F).

Discussion

The oldest-known occurrence of the Herpetogaster genus is from the lower Cambrian (Series 2, Stage 3) Chengjiang biota of China (Yang et al., 2020, 2021). The Chengjiang specimens, however, belong to H. haiyanensis, which were found in the Haiyan Lagerstätte. From the Haiyan locality, a total of eight specimens, including a juvenile specimen, were described (Yang et al., 2020). The subsequent occurrences are all representatives of H. collinsi, and include specimens from the Comet Shale, Nevada (Kimmig, Meyer & Lieberman, 2019), a possible specimen from the Parker Quarry, Vermont (Pari, Briggs & Gaines, 2022), and now the specimens reported herein from the Balang Formation (Fig. 5). The youngest representatives known to-date are also specimens of H. collinsi, recovered from the Burgess Shale (Caron, Conway Morris & Shu, 2010). While this is likely not yet a complete picture of the distribution and diversity of Herpetogaster through time and space, it suggests that the genus might have originated in Gondwana sometime around Cambrian Stage 3 and had likely spread globally, at least in the equatorial range (Fig. 1C), by Cambrian Stage 4. The current fossil record, with the oldest representatives from the Chengjiang biota (Yang et al., 2020, 2021), support the origin of the genus in Gondwana; however, the addition of the new specimens from the Balang biota also suggest that H. collinsi originated in Gondwana. Li et al. (2023) suggested Herpetogaster is the earliest-diverging cambroernid, and phylogenetically positioned at the base of the Ambulacraria, which aligns well with the origin of the genus on the timeline observed. However, unequal geographic and temporal sampling of deposits that might preserve Herpetogaster, coupled with the significant amount of fossil material and research effort on Stage 3 exceptional deposits from Gondwana, might have contributed to a biased view (sensu Whitaker & Kimmig, 2020) of the origin of the genus.

This discovery of H. collinsi in Laurentia and Gondwana around the same time (Cambrian Stage 4) implores consideration of how these sessile organisms, with no known swimming appendages or mechanisms for motility over broad distances, managed to establish themselves on the shelf of two palaeocontinents separated by thousands of kilometers of open ocean (Fig. 1C). There are many challenges that must have been overcome for this migration to have taken place, not least including the distance, but also predation, and changing water temperatures and oxygenation.

Even with an expansive fossil record and a wealth of soft-bodied organisms from the global distribution of Konservat-Lagerstätten known in the Cambrian (Muscente et al., 2017), it is rare that the same species is found in both Gondwana and Laurentia. In most cases when a Laurentian species has been proposed to occur in a Gondwanan deposit, or vice versa, it has later been revised and given a new species name, or even a new genus (Hou et al., 2017; Yang et al., 2021). However, most of the genera that are shared between the Gondwanan (e.g., Balang, Chengjiang, Emu Bay, Guanshan, and Kaili) and the Laurentian (e.g., Burgess Shale, Parker Quarry, Pioche Formation, Rockslide Formation, Sirius Passet, Spence Shale, Wheeler Formation, Marjum Formation, and Weeks Formation) biotas are arthropods. Many of these arthropods were pelagic and motile swimmers, and could have been actively seeking new habitats with new or greater resources (Legg & Vannier, 2013; Kimmig & Pratt, 2015; Robison, Babcock & Gunther, 2015; Foster & Gaines, 2016; Paterson et al., 2016; Hou et al., 2017; Lerosey-Aubril et al., 2018, 2020; Fu et al., 2019; Harper et al., 2019; Kimmig et al., 2019; Kimmig et al., 2023; Ma et al., 2020; Nanglu, Caron & Gaines, 2020; Yang et al., 2021). However, Herpetogaster, as a substrate-anchored deuterostome, requires fundamentally different considerations.

An equally plausible alternative for the palaeogeographic distribution of arthropods considers their larvae; the small size and zooplankton-like habit of the post-embryonic nauplius larval stages of arthropods could have traversed long distances by ocean currents (Müller & Walossek, 1986; Zhang & Pratt, 1993; Waloszek & Dunlop, 2002). Therefore, it follows that we should also consider the life cycle of Herpetogaster, or at least what we can infer from modern analogues. Though we have numerous fossils of the genus, much remains unknown about its ontogenesis. We can make some assumptions from modern ambulacrarian species; for example, most reproduce sexually, and many develop through a ciliated, free-swimming and feeding larval stage before settling and attachment (e.g., Thorson, 1950; Mileikovsky, 1971; Grahme & Branch, 1985; Mercier & Hamel, 2009; Wangensteen, Palacin & Turon, 2016; Gonzalez, Jiang & Lowe, 2018). Depending on taxon, this motile larval form goes by numerous names, for example, tornaria for enteropneusts, pluteus for echinoids, auricularia for holothurians, and doliolaria for crinoids, ophiuroids and holothurians. These larvae bear little anatomical resemblance to the adult forms, undergoing considerable morphological change before entering their respective adult life stages (Ettensohn, Wessel & Wray, 2004). One could suggest from the close affinity of Herpetogaster to modern invertebrate deuterostomes (Caron, Conway Morris & Shu, 2010; Nanglu et al., 2023) that it might have also had a planktonic larval stage. Some evidence for the presence of planktonic larvae has been proposed in acorn worms from the Chengjiang biota (Yang et al., 2022), thus making this a tantalizing hypothesis. As Herpetogaster has been recovered at the base of the Ambulacrarian tree in recent phylogenies (Li et al., 2023), it suggests that the last common ancestor of the clade might have already had a planktonic larval stage, or alternatively that it developed several times within the Ambulacraria.

In addition to suggestions from phylogenetic affinity, another indication of plausible larval dispersal in Herpetogaster may come from their gregarious life habit. As noted, we observed some of the H. collinsi specimens from the Balang Formation found together on a single slab (Figs. 3C and 3F), which lends support to former suggestions of a gregarious lifestyle (Caron, Conway Morris & Shu, 2010; Kimmig, Meyer & Lieberman, 2019). A planktonic larval stage is common in many other modern gregarious marine invertebrates (Pechenik, 1999; Toonen & Pawlik, 2001), with dispersal and global connectivity well-modeled from broadcast spawning of corals (Wood et al., 2014), and has been hypothesized for more ancient gregarious taxa as well (Cortijo et al., 2015; Schiffbauer et al., 2016). The gregarious lifestyle of H. collinsi may indicate that it, too, was a broadcast spawner, where fertilization and development of the offspring happened externally, promoting wide dispersal of embryos and larvae and resulting in a broad distribution of offspring. The presence of a larval stage would also explain the relative longevity of the genus and species, as organisms with larval stages are more resistant to extinction as they may have multiple distant populations that can survive even when a local population is subjected to disaster (Hansen, 1978, 1980; Jablonski, 1986). While more evidence is necessary to confirm our interpretation, we observed a single millimetric ovoid structure in close proximity to an adult Herpetogaster specimen (Fig. 3A). From hints of a lateral band extending around this ovoid, we offer a tentative suggestion that it may be a possible larva (Fig. 3B), as it is comparable in size and form to those of modern acorn worms or other echinoderms (e.g., Yang et al., 2022). This potential larva is understandably speculative, as it does not preserve enough details to be definitively identified, and other examples within this deposit have not yet been identified. However, previous authors have argued that larvae (sensu Hickman, 1999; Haug, 2018) were likely the key to the worldwide distribution of deuterostomes from Burgess Shale-type biotas (e.g., Han, Zhang & Liu, 2008; Yang et al., 2022), and larvae are already known to play a vital role in Cambrian arthropod development and dispersal (e.g., Liu et al., 2016; Lerosey-Aubril & Laibl, 2021; Laibl, Saleh & Pérez-Peris, 2023)—both of which urge the continued search for more possible larval fossils from the Balang.

Larval fossils are rare in the fossil record compared to later developmental stages based on both biological and taphonomic factors (e.g., Laibl, Saleh & Pérez-Peris, 2023). Furthermore, their small size and often unassuming nature can lead to specimens being overlooked or left behind in the field (Whitaker & Kimmig, 2020). The likely soft-bodied nature of Herpetogaster larvae is an additional issue in recovering specimens, as they decompose quicker than biomineralized larvae of arthropods. However, the Balang Formation, as well as other Cambrian deposits that contain soft-bodied fossils, have the preservation potential for these important fossils, and it is important to pay attention to microscopic carbonaceous materials as they might actually represent larvae.

We suggest that Herpetogaster having a planktonic larval stage and reproduction through broadcast spawning would have permitted dispersal of this genus over long distances, enabling them to establish a foothold in both Laurentia and Gondwana and providing a logical explanation for the sum of our observations. The question that remains is whether planktonic larvae were shared across the Herpetogaster genus, or if they developed later in H. collinsi. Testing this hypothesis, however, will require further discoveries of H. haiyanensis outside of the Chengjiang region or additional species elsewhere in the Cambrian. Continued investigation of the Balang biota is necessary to confirm or adapt our provisional inferences based on the presence of the tentative but promising tornaria-like larva.

Conclusions

The discovery of H. collinsi from the Balang Formation of China represents the first report of this species from Gondwana. Most specimens are complete and the detailed preservation, which required exceptional depositional circumstances, makes an assignment to the species unquestionable. However, the presence of the same sessile species in Laurentia and Gondwana poses the question as to how these organisms managed to colonize two distant parts of the world. We suggest that the most likely scenario sees Herpetogaster with a planktonic larval stage and reproduction through broadcast spawning, which would have permitted dispersal over long distances and enabled H. collinsi to establish a foothold in both Laurentia and Gondwana. As Herpetogaster is already known from Cambrian Stage 3 in Gondwana, it would suggest a possible origination of the genus there, but the limited record of the genus at that time does not yet allow for a definite conclusion.

Supplemental Information

Supplemental Information 1 Measurements of the Balang Formation specimens.

Click here for additional data file.

Supplemental Information 2 10keV SEM micrograph and SEM-EDS elemental maps of Herpetogaster collinsi from the Balang Formation of China.

(A–G) YKLP 14573. (A) Picture of the specimen indicating the analyzed area (dashed rectangle). (B) Detailed view of the analyzed area. (C–G) SEM-EDS elemental maps of Ca, Fe, C, P, S, respectively. (H–N) YKLP 14583. (H) Picture of the specimen indicating the analyzed area (dashed rectangle). (I) Detailed view of the analyzed area. (J–N) SEM-EDS elemental maps of Ca, Fe, C, P, S, respectively. Scale bars: (A, H) 5mm; (C–G, J–N) 1 mm.

Click here for additional data file.

This article is a contribution to IGCP668, Equatorial Gondwanan History and Early Palaeozoic Evolutionary Dynamics. We thank Jean-Bernard Caron (ROM) for pictures of the holotype of Herpetogaster collinsi and Derek Briggs (YPM) for pictures of the Parker Quarry? Herpetogaster specimen. We thank the reviewers Farid Saleh, Steve Pates and Jeffrey Thompson and the editor Paula Bona for their constructive suggestions.

Additional Information and Declarations

Competing Interests

Author Contributions

Data Availability

The authors declare that they have no competing interests.

Xianfeng Yang conceived and designed the experiments, performed the experiments, analyzed the data, prepared figures and/or tables, authored or reviewed drafts of the article, and approved the final draft.

Julien Kimmig conceived and designed the experiments, performed the experiments, analyzed the data, prepared figures and/or tables, authored or reviewed drafts of the article, and approved the final draft.

James D. Schiffbauer conceived and designed the experiments, performed the experiments, analyzed the data, prepared figures and/or tables, authored or reviewed drafts of the article, and approved the final draft.

Shanchi Peng analyzed the data, authored or reviewed drafts of the article, and approved the final draft.

The following information was supplied regarding data availability:

The raw measurements are available in the Supplemental File.

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
