# Peer review of "Herpetogaster collinsi from the Cambrian of China elucidates the dispersal and palaeogeographic distribution of early deuterostomes and the origin of the ambulacrarian larva"

_PeerJ, doi:10.7717/peerj.16385_

## Round 0.1 · original submission · Minor Revisions

Please improve Figure 1 as needed and follow other minor comments highlighted by me in the pdf.

·

Basic reporting

The paper is well written, clear, and scientifically sound. It could benefit from some additional references, as indicated in my detailed review below. And the captions of the figures must be expanded as they are too brief in their current form.

Experimental design

The paper fits within the scope of the journal. The science is well-defined, relevant, and ethically and rigorously conducted. The data presented is sufficient to replicate the study.

Validity of the findings

The data presented is sound, and the conclusions are well stated and clear. The impact of the paper is understated in some sections (e.g., the abstract), and I suggest some minor edits below to make the paper stand out more.

Additional comments

Yang et al. describe a new material of Herpetogaster collinsi and contextualize it within an ecological/evolutionary framework. The paper is well-written, and the science is original, convincing, and fills a knowledge gap. I enjoyed reading the paper as it is clear and straightforward. I recommend publication following minor to moderate revisions.

My main concerns are related to the SEM EDX analyses. Figure 4 does not clearly show anything, and I believe one of the reasons is that the authors used a 20keV accelerating voltage. At this voltage, light elements such as carbon may get hidden, and the beam could penetrate the sample, resulting in an averaged map of the fossil and the underlying matrix. Lowering the voltage (to 10) might lead to a better identification of light elements such as carbon on the sample's surface. Although some resolution might be lost, I believe it will be more informative than the 20keV data presented here.

The authors also mention that “Organic carbon removal/dispersion could have continued through diagenesis”, which, in principle, I agree with. However, it is equally plausible that modern episodes of weathering, such as precipitation, could have leached organic material (e.g., Saleh et al., 2021, Earth Science Reviews). Without clear analyses to distinguish between these processes, it will be difficult to confidently conclude that all the loss occurred during diagenesis.

Another main concern is that the captions of the figures are very brief and feel rushed. To improve the manuscript, I suggest expanding the captions by adding a description to each of the panels of the figures. This is crucial and will make the manuscript stand out more.

Additionally, I believe the manuscript can stand out even more with some minor edits to the text, starting from the abstract. For instance, instead of concluding the abstract with a general sentence that does not provide much insight into the new findings, consider replacing it with a sentence directly linked to the science presented herein (e.g., a sentence about the motile larval stage?).

Lastly, the manuscript would benefit from additional references, as some sections are well-referenced while others lack proper referencing. For example, lines 274-278 would benefit from some references, as this section is crucial to the core of the manuscript. Also, in lines 58-59, I suggest adding Saleh et al., 2020, Earth and Planetary Science Letters, as it contains a relevant database of generic occurrences of Walcott and Chengjiang.

In summary, I would like to congratulate the authors on the paper. With some edits to the text and figures and potential new analyses, I am confident that it can make a significant contribution to the field.

Farid

·

Basic reporting

Clear and unambiguous professional English is used throughout. The article conforms to professional standards of courtesy and expression.
Sufficient background is provided and the references are sufficient.
The article is well structured and conforms to an acceptable format.
Figures are clear, relevant, and appropriately described and labelled.
---The numbers in Fig. 1C are not clear to me, could they be increased in size?
The submission is self-contained and presents results relevant to the discussion.

Experimental design

The research is original and fits within the Aims and Scope of PeerJ
It is clear how the research fills a knowledge gap.
The investigation of the fossil material and its preservation is rigorous. While the discussion is speculative it is clearly highlighted as such.
Methods are well described and could be replicated with the relevant equipment.

Validity of the findings

All underlying data have been provided.
Conclusions are clearly stated and link to the original question.

Additional comments

Line 78 – ‘recent flume experiments have shown that soft-bodied animals can hypothetically be transported over tens of kilometers by current flow’. The authors state that this is an improbable explanation, and I agree. However given that it is the only alternative hypothesis considered, it is worth expanding on a little further. Could the authors add one or two more sentences, perhaps considering the presence of other continents (e.g. Siberia, Australia in Fig. 1C) and vast distance (>10s of km) between South China and Laurentia, to more fully justify this sentence?
Line 237 ff. – I am not sure about the origination and dispersal proposed here. Given the scarcity of Series 2 Stage 3 deposits outside South China where Herpetogaster might be preserved and discovered, it is unclear to me whether the genus originated there or if instead we find the oldest material there due to the significant amount of fossil material and research effort on Stage 3 exceptional deposits. I suggest the authors at least add a statement acknowledging the unequal geographic and temporal sampling of deposits that might preserve Herpetogaster and how this influences their dispersal hypothesis.
Lines 274-6 – ‘we can make some assumptions from modern ambulacrarian species…’ – can the authors provide some references for this sentence?
Line 283 – this study (Yang et al. 2022) is a preprint. If it is not published before acceptance of this manuscript I would suggest softening the language here to, from ‘…larvae has been found in acorn worms…’ to ‘…larvae has been proposed in acorn worms…’ to reflect this.
Further comment – given the range of deposits from which Herpetogaster collinsi has now been recovered from (and the range of ages) can the authors comment on the range of environments that this species was able to colonize and/or its relative longevity compared to other Cambrian species?
Further comment 2 – the authors clearly highlight that their identification of a larva is speculative (line 303) and that more work is required to demonstrate a larval stage in H. collinsi (e.g. paragraph beginning line 268, sentence beginning line 317). Could the authors expand on what investigations are required to confirm/adapt/overturn their provisional inferences that H. collinsi possessed a planktonic larval stage? Comparison to recent work by Laibl on trilobites (e.g. Laibl et al. 2023 Palaeo3 https://doi.org/10.1016/j.palaeo.2023.111403) might be useful? Setting out a workflow or threshold of evidence required to be confident of planktonic larvae in H. collinsi would make the finale of the discussion more compelling.

Reviewer 3 ·

Basic reporting

The manuscript by Yang et al. provides a nice description of a new fossil cambroernid belonging to the genus Herpetogaster and a discussion of its implications for biogeographic dispersal during the Cambrian. The manuscript is well-written, easy and enjoyable to read and the figures are clear and appropriate. I have made a few minor comments on the attached manuscript, and have listed more substantive comments herein.

Experimental design

Experimental design is good. No other comments.

Validity of the findings

My main points are:
1) I think the introduction could use a bit more text introducing the cambroernids, in terms of their general bodyplan construction, their diversity etc. Many readers, especially those who don’t work in the Cambrian, will not be familiar with these animals
2) Given the importance placed on the interpretation that these taxa belong to the same species as those reported from the Burgess Shale and Pioche Formation, I think the manuscript could benefit with a more detailed discussion (e.g. few lines in the Remarks section), justifying why these specimens are classified as members of the same species. You highlight that this is because of similar morphology, but I think this should be fleshed out in more detail, as opposed to simply listing the morphological structures that are similar.
3) I think you’re missing a trick with this manuscript in terms of what you choose to highlight early on. I think you make a good case that this animal probably had a larval stage which was broadly distributed and the implications for the ambulacrarian MRCA. Ambulacrarian larval origins are a topic of interest to neontologists, so I think you should highlight these implications in the abstract, and probably even title. I think its cool that the fossil may have a role to play in this discussion!

Additional comments

None.

Annotated reviews are not available for download in order to protect the identity of reviewers who chose to remain anonymous.

---

## Round 0.2 · accepted · Accept

Authors have addressed all of the reviewers' comments. This manuscript is ready for publication.